# The Impact of Salt Accumulation on the Growth of Duckweed in a Continuous System for Pig Manure Treatment

**DOI:** 10.3390/plants11233189

**Published:** 2022-11-22

**Authors:** Marie Lambert, Reindert Devlamynck, Marcella Fernandes de Souza, Jan Leenknegt, Katleen Raes, Mia Eeckhout, Erik Meers

**Affiliations:** 1Provincial Research and Advice Centre for Agriculture and Horticulture (Inagro vzw), Ieperseweg 87, 8800 Roeselare-Beitem, Belgium; 2Lab for Bioresource Recovery, Department of Green Chemistry and Technology, Ghent University, Coupure Links 653, 9000 Ghent, Belgium; 3Research Unit VEG-i-TEC, Department of Food Technology, Safety and Health, Ghent University, St-Martems Latemlaan 2B, 8500 Kortrijk, Belgium; 4Research Unit of Cereal and Feed Technology, Department of Food Technology, Safety and Health, Ghent University, Valentin Vaerwyckweg 1, 9000 Ghent, Belgium

**Keywords:** Lemnacea, alternative protein, agricultural wastewater, water recovery, accumulation, continuous systems

## Abstract

Duckweed (*Lemna*) is a possible solution for the treatment of aqueous waste streams and the simultaneous provision of protein-rich biomass. Nitrification-Denitrification effluent (NDNE) from pig manure treatment has been previously used as a growing medium for duckweed. This study investigated the use of a continuous duckweed cultivation system to treat NDNE as a stand-alone technology. For this purpose, a system with a continuous supply of waste streams from the pig manure treatment, continuous biomass production, and continuous discharge that meets the legal standards in Flanders (Belgium) was simulated for a 175-day growing season. In this simulation, salt accumulation was taken into account. To prevent accumulating salts from reaching a toxic concentration and consequently inhibiting growth, the cultivation system must be buffered, which can be achieved by altering the depth of the system. To determine the minimum depth of such a system, a tray experiment was set up. For that, salt accumulation data obtained from previous research were used for simulating systems with different pond depths. It was found that a depth of at least 1 m is needed to prevent a significant relative growth inhibition at the end of the growing season compared to the start. This implies a high water consumption (5–10 times more than maize). As a response, a second cultivation system was investigated for the use of more concentrated NDNE. For this purpose, salt tolerance experiments were conducted on synthetic and biological media. Surprisingly, it was observed that duckweed grows better on diluted NDNE (to 75% NDNE, or EC of 8 mS/cm) than on a synthetic medium (EC of 1.5 mS/cm), indicating the potential of such a system.

## 1. Introduction

Manure and agricultural wastewater treatment is a worldwide problem [1]. In some regions with intensive agriculture, such as Flanders (Belgium), manure application on land is limited and therefore its surplus needs to be treated to prevent eutrophication [2]. A considerable amount of nitrogen and phosphorus is therefore removed during the treatment of surplus manure and cannot be used for crop production [3]. At the same time, there is a significant import of fertilizers and proteins in these regions. Therefore, using this surplus manure can help to close the nutrient loop in the region.

A potential solution to close this cycle can be the cultivation of duckweed on wastewater. Duckweed is a general name for plants that belong to the family of Lemnaceae [4]. These small floating macrophytes can be found all over the world and belong to the most rapidly growing Angiosperms, following a quasi-exponential growth rate [5]. In Europe (Ireland), production rates of 37.89 tonnes of dry duckweed biomass have been reported [6]. However, higher productivity levels of 68 tonnes DW ha^−1^ year^−1^ have also been reported [7]. Besides productivity, the primary advantage of duckweed is its high crude protein content of around 35% [8], and up to 45% DW [9]. Additionally, its moderate amount of fibres (5–15% on DW) makes it readily digestible for monogastric animals and many fishes [10,11].

Similar to the wetland technology developed to process pig manure [12], duckweed forms a floating mat that can remove N and P from wastewater. The advantage of duckweed in comparison to wetlands is the simultaneous provision of proteins for livestock production [13]. In this way, nutrients are recycled into a key feed ingredient. Duckweed has been shown to grow on dairy wastewater [14,15], pig manure wastewater [7,16] and aquaculture effluent [17,18].

Figure 1 presents an illustration of a manure treatment process and how duckweed can be added to that process. The first part of this scheme (up to NDNE) has already been applied at a manure treatment plant. The second part, the duckweed cultivation system, has not yet been applied in practice but was simulated in this study. This system was filled with a certain amount of NDNE and liquid fraction of pig manure (LF), and was diluted with rain water (RW) at the beginning of the growing season. Subsequently, a continuous supply of NDNE and LF was added so that the added N and P concentration was equal to the treatment capacity of the system, which equaled 1107 ± 715 mg/m²/d and 149 ± 150 mg/m²/d, respectively [19]. In Flanders (Belgium), it is only feasible to grow duckweed for around 175 days each year, as was determined by Devlamynck et al. [13]. During this period, a continuous flow of dischargeable water is generated at the end of the system. After the growing season, the duckweed tank can be gradually emptied for the remaining days of the year. 

In the duckweed cultivation system, as presented in Figure 1, nutrients can accumulate over time when their incoming concentration is higher than the concentration that can be removed by the system [19]. It is, however, possible to reduce the accumulation rate by adjusting the system’s volume or, in other words, the pond’s depth. The deeper the pond, the lower the concentration of accumulating nutrients will be after one growing season. Yet, the depth must be minimized in order to reduce the water consumption of the system.

Therefore, this study investigated the possibility of using a continuous duckweed cultivation system to treat NDNE as a stand-alone approach (as represented in Figure 1). For this purpose, a system with a continuous supply of waste streams and a continuous discharge that meets the legal standards in Flanders (Belgium) was simulated for a 175-day growing season. Next, a tray experiment was carried out to determine the minimum depth of such a system. Furthermore, the impact of dilution on the suitability of NDNE as a growing medium for duckweed was also assessed. From salt tolerance experiments carried out by Landolt et al., an electrical conductivity (EC) of 10.9 mS/cm has been proposed as a maximal salt stress level for duckweed [9]. This indicates that undiluted NDNE cannot be used for duckweed cultivation as variations in EC between 7 and 24 mS/cm were observed [20]. However, EC does not distinguish between cations and anions or the share of beneficial and harmful elements. For example, Walsh et al., recently highlighted the Ca-to-Mg ratio as an extremely important parameter for duckweed growth [21]. Therefore, this study also investigated if the ratio of anions can play a role in the salt tolerance of duckweed.

## 2. Results and Discussion

### 2.1. Simulation of a Continuous Duckweed Cultivation System (Tray Experiment 1)

In a continuous system, the accumulation of ballast salts can occur [19], which can result in toxic salt levels and lead to a culture crash. Therefore, experiments mimicking continuous systems for obtaining more accurate results and assessing their water consumption are needed. In this study, a continuous system with a continuous supply of NDNE and LF and a continuous discharge that meets the discharge standards in Flanders (Belgium) that works for 175 days was simulated. The medium composition was calculated for systems with different depths (0.4 m, 0.7 m and 1 m). After simulating the system, a tray experiment was carried out with the calculated compositions. The relative growth (RGR) and chlorophyll inhibitions results, obtained from this experiment, are shown in Figure 2.

It was observed that the RGR inhibition of the continuous systems with a depth of 0.7 m or 0.4 m was significantly higher at the end of the growing season compared to the beginning of the season (*p* = 3.98 × 10^−3^ and 1.28 × 10^−11^, respectively). Regarding chlorophyll inhibition, it was shown that the reference medium had a significantly higher chlorophyll content compared to the other treatments (*p* < 8.99 × 10^−7^). When comparing the chlorophyll concentrations at the start of the growing season with those at the end of the growing season, only the system with a depth of 0.7 m showed a significant inhibition (*p* = 3.56 × 10^−2^). For the 0.4 m system, the duckweed obtained in the pre-cultivation exhibited a significant chlorophyll inhibition (*p* = 3.56 × 10^−2^) but seemed to recover during the cultivation step.

As expected, the RGR of duckweed grown in a deeper system was less inhibited after 175 days than when it was growing in a shallower system. Accordingly, the deeper the system, or the higher the buffering capacity for the accumulating salts that enter the system, the lower the inhibition after one growing season. Roughly, a similar trend was shown for chlorophyll inhibition. The observation that the reference medium had a higher chlorophyll content can be explained by the low N and P concentrations of the wastewater-based growing media. Additionally, the pH of the reference medium was closer to the optimal pH for duckweed growth (Table A3) [9].

It can be concluded that, when duckweed is grown in a system with a continuous supply of NDNE and LF and a continuous discharge that meets the legal standards in Flanders (Belgium), a buffering capacity of at least 1 m depth is recommended. This would ensure that the relative growth inhibition would not be significantly higher at the end of the growing season compared to the start of the cultivation. However, in practice, this would mean that the continuous cultivation of duckweed would consume a significant amount of water. A buffering capacity of at least 1 m depth means that, for the cultivation of one hectare for one growing season, 10,000 m³ water is needed. This is significantly higher compared to other crops such as maize, which has an irrigation water requirement of 900 to 1750 m³/ha/yr for optimal growth [22]. In reality, the water consumption itself may be even higher than calculated in this simulation, as the experiment did not take into account the naturally occurring evaporation of water. 

The high water consumption is a result of the toxicity that occurs due to the accumulation of nutrients because of the continuous addition of biological waste streams. According to our simulation, this toxicity is mainly determined by potassium, since this element exceeds the toxicity limit according to Landolt et al., in the shortest time (see Table A3). However, if one considers the actual measured values before and after cultivation (Table A1 and Table A2), it can be observed that the toxicity limit of potassium (2000 mg/L) was not exceeded. None of the growing media, neither before nor after cultivation, exceeded the toxicity limit for any element according to Landolt and Frick et al. [9,23]. Nevertheless, some elements, such as K and Cl, lie further outside their optimum range when a shallower system is simulated, which may be the reason for increased stress and therefore growth and chlorophyll inhibition.

Next to the high K concentration, a possible driving factor determining toxicity may be an adverse balance of nutrients. For example, it is known that a ratio which favors Mg over Ca negatively affects *L. minor* growth and its photosynthetic yield [21]. Therefore, a Ca:Mg ratio of 1:1.6 or greater is recommended for *L. minor* growth. In Figure 3 it is shown that, during cultivation, the Mg/Ca ratio of the reference and start medium was significantly lower than the maximum ratio of 1.6 (*p* = 8.33 × 10^−5^ and 1.42 × 10^−7^, respectively). For the least buffered medium (0.4 m) it was shown that the Mg/Ca ratio was significantly higher than this maximum ratio (*p* = 9.16 × 10^−5^).

The shallower the system or the less buffering capacity the system has for accumulating salts, the higher the growth inhibition will be, and also, the more the nutrient uptake will decrease and will even become negative (Table A7 and Table A8). A negative nutrient uptake indicates leaching, which is a clear signal of plant stress. A negative removal was observed for the elements Ca, P and Mn. This is an issue, especially for phosphorous, as the system was created/simulated in such a way that the added P concentrations should equal the removed P concentration. These measured nutrient removals also do not match those published by Devlamynck et al. [19], which were used to calculate the needed medium concentrations for the simulation. In addition, a constant removal was used to calculate all simulations for each nutrient, whereas we can see here that, for most nutrients, removal increases with increasing concentration. Even though more investigations are needed for continuous systems, this study indicates the importance of taking into account the salt accumulation in such setups.

Besides the high water consumption, a big disadvantage of the simulated system is that the growth medium must be discharged after 175 days, at the end of the growing season. Discharging this quantity of growth medium involves some practical problems, as the medium is too salty to be discharged into nature as a single flush. A possible solution is to opt for 175 days of duckweed growth in one year and then to gradually discharge for the rest of the year until the system is empty on the 365th day, as illustrated in Figure 1.

### 2.2. Duckweed Cultivation on NDNE (Tray Experiments 2 and 3)

#### 2.2.1. NDNE as a Growing Medium

The simulation discussed in the previous section focuses on a system that treats NDNE and LF till a dischargeable effluent is obtained. However, it might be more interesting to treat a higher concentration of NDNE in a continuous system to achieve a higher biomass production and to reduce the water consumption needed for dilution. The purpose of the system is then changed from ‘treating a waste stream as a stand-alone system to a dischargeable effluent’ to ‘production of protein-rich biomass with recycled nutrients’. In order to still obtain a dischargeable effluent at the end, the system can be combined with a constructed wetland using reedbeds as a final purification step.

To assess the maximum accepted waste stream concentration to minimize the needed dilution, a second tray experiment was done. Nine different dilutions/treatments of NDNE were tested (Table A2). It was investigated if the cultivation of duckweed on a biological medium (after dilution, addition of a salt solution or evaporation) resulted in better or worse growth compared to cultivation on a synthetic nutrient medium. In Figure 4, the RGR inhibition and the chlorophyll inhibition of the duckweed grown on different media are shown.

Surprisingly, the different dilutions and the undiluted NDNE showed no RGR inhibition compared to the reference medium. Even for the treatments where the EC was artificially increased there was only a significantly higher RGR inhibition after evaporation until an EC of 11.8 mS/cm was obtained (*p* = 2.38 × 10^—9^). However, for chlorophyll inhibition, a different trend was shown. There was a significantly higher chlorophyll inhibition for the undiluted NDNE and for most of the treatments where the EC was increased, indicating plant stress. For the diluted NDNE to an EC of 4, 6, or 8 mS/cm, the chlorophyll inhibitions were significantly lower than the reference (*p* = 1.48 × 10^−6^, 1.77 × 10^−12^ and 1.28 × 10^−13^, respectively). The RGR and chlorophyll inhibitions of these dilutions were even below 0%.

From the results shown in Figure 4, we can conclude that duckweed grew better on the diluted NDNE than on the reference medium. This is surprising, as the N medium is described in the literature as one of the best nutrient media used to support the fast growth of duckweed [24]. A possible explanation lies in the presence of organic components in the used waste streams. Organic matter contains humic substances that can be divided into three classes: fulvic acids (FA), humic acid (HA), and humin [25]. Humic substances might increase the uptake of both macro and micronutrients, such as N, P, K, Fe and Zn [26]. Additionally, they might also reduce the plant uptake of certain toxic metal ions, like Cd [27]. Thus, one might reason that the application of humic substances could improve plant response to salinity. However, in the literature, opinions are very much divided on this hypothesis. Liu et al., studied the influence of HA on the salt tolerance of hydroponically grown creeping bentgrass. They found out that, in general, the application of HA did not improve the salinity tolerance of the plant [28]. However, different findings were observed for other crops. Ghulam et al., studied the influence of HA on salt tolerance and nutrient uptake in wheat. They found out that a combined application of K and HA was promising for increasing wheat salt tolerance and nutrient uptake [29]. As both studies show very different results and worked with other crops than duckweed, it is difficult to make any concrete conclusions other than a possible indication that duckweed might behave like wheat and have higher salt tolerance in the presence of humic substances. Another explanation lies in the form in which nitrogen is present in the growth medium. Ammonium is the preferred nitrogen source of duckweed [30] and NDNE has more nitrogen present in the form of ammonium compared to the reference medium (Table A1), in which all the nitrogen is present in the form of nitrate.

In this study, it was shown that the duckweed grew better on the diluted NDNE media than on the reference medium. However, it should be considered that only data on growth and chlorophyll concentration was studied here. Osmotic stress can cause a reduction in protein concentration. Therefore, an important next step to this study should be biomass quality assessment of the duckweed grown on these different media.

#### 2.2.2. Salt Tolerance of Duckweed

From the previous tray experiment, it seemed that duckweed is less salt-sensitive when grown on a biological medium compared to when grown on a synthetic medium. In order to better understand the salt tolerance of duckweed, an additional tray experiment was done. In this third experiment, a dose-response curve of *Lemna minor* was determined after adding different concentrations of NaCl to the synthetic reference medium. This approach is similar to how toxicity tests are usually done in literature [31]. However, this may not be the best way to determine salt tolerance as it is also important to take into account the ratio of certain elements. This was already demonstrated for the Mg/Ca ratio for example. Therefore, the results of this tray experiment were compared with another experiment where the same amount of Na^+^ was added to a synthetic medium by adding a mixture of NaCl, Na_2_SO_4_ and K_2_SO_4_. This mixture had an SO_4_^2−^/Cl^−^ ratio of 0.25, similar to the ratio measured in NDNE in our previous research.

In Figure 5 both the RGR- and the chlorophyll inhibition are plotted as a function of the EC and the concentration of sodium in the mixture. It is shown that the EC is lower when Na^+^ is added as NaCl than when the same concentration of Na^+^ is added as a combination of NaCl, Na_2_SO_4_ and K_2_SO_4_. This is because SO_4_^2−^ has twice as many charges as Cl^−^ and will therefore have a greater influence on the conductivity. Next, it was shown that the RGR inhibition was significantly higher at a certain EC when only NaCl was added compared to when a combination of NaCl, Na_2_SO_4_ and K_2_SO_4_ was added. A significant higher growth inhibition was observed after adding 46.6 mM Cl^−^ to the medium (up to EC = 6.51–6.65 mS/cm), compared to the reference medium (0 mM NaCl added—EC = 1.5 mS/cm) (*p* = 2 × 10^−3^ for pre-cultivation, *p* < 1.08 × 10^−9^ for cultivation).

Remarkably, when adding a combination of Cl^−^ and SO_4_^2−^, a significantly higher tolerance was observed in terms of growth inhibition in function of the EC, as total growth inhibition was only obtained after the addition of 37.3 mM Cl^−^ and 9.32 mM SO_4_^2−^ to the medium (up to EC = 8.18–8.43 mS/cm) (*p* = 4.88 × 10^−7^ for cultivation). Presumably, total inhibition could have occurred even at lower concentrations than 46.6 mM NaCl. Therefore, it may be assumed that, in a synthetic medium, the duckweed was less inhibited when an equal amount of Cl^−^/SO_4_^2−^ was added than when only Cl^−^ was added. 

The same trends are shown in Figure 5C,D for chlorophyll inhibition. However, it is possible to see significant differences at lower EC values or after the addition of lower Na^+^ concentrations. A significantly higher chlorophyll inhibition was observed for pre-cultivation after adding 18 mM Cl^−^ to the medium (up to EC = 3.5 mS/cm), compared to the reference medium (0 mM NaCl added—EC = 1.5 mS/cm) (*p* = 3.24 × 10^−4^ for pre-cult). Remarkably, when adding a combination of Cl^−^ and SO_4_^2−^ ions, a significantly higher tolerance was observed for the pre-cultivation, in terms of both growth inhibition in function of the EC and the Na^+^ concentration. However, total chlorophyll inhibition was only obtained for the pre-cultivation after the addition of 25.6 mM Cl^−^ and 6.4 mM SO_4_^2−^ to the medium (up to EC = 6.25 mS/cm) (*p* = 2.19 × 10^−3^). For the cultivation step, there was, for both experiments, a significant chlorophyll inhibition after the addition of 32 mM Na^+^ (up to EC = 4.96–5.05 for Cl^−^ addition; EC = 6.25–6.44 for Cl^−^ and SO_4_^2−^ addition) (*p* = 1.02 × 10^−5^ for Cl^−^ addition; *p* = 5.63 × 10^−5^ for Cl^−^ and SO_4_^2−^ addition). The chlorophyll content of a plant can be used as a measure to assess oxidative damage in salt treatments. Oxidative stress is usual before a decrease in growth or die-off. In this sense, it is expected that chlorophyll inhibition already occurs at lower concentrations and EC values.

To conclude, in this experiment, duckweed was less inhibited when an equal amount of Cl^−^/SO_4_^2−^ was added than when only Cl^−^ was added. This proves that the composition of the anions plays a role in the salt tolerance of duckweed. As a result, the observation from the previous experiment that duckweed is less salt-sensitive when grown on a biological medium may be partly explained by the composition of the anions in this medium.

#### 2.2.3. Variation in NDNE

From the second tray test, it could be concluded that the best medium for duckweed consists of a mixture of 75% NDNE and 25% demineralized water (EC = 8), both in terms of growth and chlorophyll concentration. However, the composition of the NDNE is not constant over time. In order to demonstrate that variation, the EC of NDNE was monitored in situ in a treatment facility. Over the same period, also the precipitation was monitored to calculate the dilution of the treatment system over its retention period (36 days). 

Figure 6 shows that there is variation in the EC of NDNE over time. In fact, for 49% of the time, the measured EC was higher than 10.9 mS/cm, which is the maximal EC for duckweed survival according to Landolt et al. [9].

It was investigated whether the precipitation influenced the variation of the EC, as the rainwater that enters the system may dilute the stream and thus lower the EC, causing a higher variation. In this way, a lower EC could be expected in periods with high precipitation. To visualise this impact in the graph, a corrected EC was calculated by assuming that the rainwater entering the NDNE has an EC of 0. This leads to an underestimation but clearly shows that rainwater has no to little effect on the variation of the EC on NDNE.

More likely, the variable concentration of anions and cations in the NDNE is mainly caused by variations in the inlet stream and other process parameters. In this case, the monitored pig manure processing treatment has an inlet stream that consists of a combination of manure from fattening pigs and sows. The manure of sows is worth more for fertilisation and is preferred in times of high demand. Thus, around 10/05, the manure of sows was all diverted for fertilizing the lands. Hence, the manure from fattening pigs was fed to the treatment. The latter is thicker and has a higher conductivity. As a result, we can observe increasing conductivity from then on.

To conclude, the variation of the source material determines to a large extent the variation in the final growth medium. This variation can be reduced by the installation of buffering or storage lagoons, which would allow for a more stable effluent composition and an easier formulation of an adequate medium for duckweed growth on NDNE.

## 3. Materials and Methods

### 3.1. Tray Experiments

#### 3.1.1. Experimental Conditions

Duckweed growth experiments were executed in PET containers (0.266 × 0.165 × 0.119 m) on a growth rack under laboratory conditions, as shown in Figure 7. The used containers were opaque to avoid light penetration through the walls and hence inhibit algae growth [32]. The total tray volume was 5.2 L, the surface area was 438.9 cm² and the trays were filled with 3 kg of growth medium for each experiment. The growth rack consisted of two operative levels. Each level could accommodate 10 containers, adding up to a total of maximally 20 containers per experiment.

Light was provided in a 16:8 h light-darkness regime by 4 parallel TL-light (TLD 36 w/86, Philips, the Netherlands) per level. Light intensity (PPFD, or photosynthetic photon flux density) ranged between 110–150 µmol/m²/s and was measured at the respective duckweed mat level in the trays, in the centre of each tray. The simulation experiment took place in a climate chamber with a temperature of 25 ± 1 °C and air humidity of 70 ± 2%. The second tray experiment took place in the lab at a room temperature of 23 ± 2 °C.

To compensate for light asymmetry, the trays that were filled with the reference medium were placed in the centre of the rack. In this way, the reference was favoured over the other treatments. For the second experiment, where different dilutions and treatments of NDNE were used as a growing medium, a rotation scheme was designed to compensate for light asymmetry. A 5-period rotation was applied to keep each container in each row for the same time on each position. Besides, the water level was weight-adjusted with demineralized water to compensate for evaporation, solution mixing and to counter possible heat effects.

Plant material was sourced from a natural pond in Rumbeke-Beitem, Belgium. Visual determination according to [33] clarified that duckweed plants belonged to the *Lemna minor* species. The identification of the duckweed species was performed using molecular barcoding based on plastidic markers prior to the experiment [34].

Plant density was selected based on the work of Monette et al., to ensure total coverage of the water surface to minimize algae growth [35]. Since the overall goal was to obtain maximal biomass production, rather than maximal relative growth rate, a relatively high initial density of 30 g fresh weight (FW) was inoculated in the trays. Therefore, the density at the start of each cultivation was equal to 683.53 g/m².

#### 3.1.2. Experimental Design

Three experiments were conducted where the dose-response curve of *Lemna minor* was determined. In the first experiment, waste streams from a pig manure treatment process at a pig farm in Pittem, Flanders (Belgium) were used to make the growth media. In the second experiment, NDNE from another pig manure treatment process in Flanders (Belgium) at IVACO, Eernegem was used. The latter was also the same manure treatment plant on which the EC of the NDNE was monitored. In the third experiment, duckweed was grown on a synthetic medium to which different concentrations of salt solutions were added.

The reference synthetic nutrient medium used in all experiments was the N medium as described in the ISCDRA newsletter [24]. The reference article for this medium was written by Appenroth et al. and these researchers stated that they “never found a nutrient media that supported a faster growth of duckweed than this one” [36]. The N medium was prepared as concentrated stock solutions (Table A1), of which 5 mL was taken for each litre of growth solution.

For each experiment, the pH of the reference medium was adjusted to 6–6.5 with 0.1 M NaOH. The other growth media used in the different experiments were also adjusted to the same pH, by adding either NaOH or HCl. For the first experiment, the pH of the growth media, other than the reference, was adjusted to the same pH as the start medium, which was 8.1 ± 1.

##### Tray Experiment 1: Simulation of a Continuous Duckweed Cultivation System

The growing medium of a continuous system with a depth of 1 m, 0.7 m or 0.4 m was simulated at the beginning and at the end of the growing season (175 days). In this tray experiment the depth of the simulated continuous system was determined. It should be stressed that the parameter of depth is a simulation using the assumption of a fixed nutrient accumulation during a growing period of 175 days. The experiment could also be interpreted as a pond with a fixed depth and where different periods of growth would be simulated if the pond would be placed in a region with a more suitable climate. For example, if the depth is fixed to 1 m, the treatments originally described as: ‘1 m’, ‘0.7 m’ and ‘0.4 m’ would then be described as: ‘after 175 days’, ‘250 days’ and ‘438 days’, as calculated with the same simulation model using a non-linear solver technique.

To prepare the different growth media for this experiment, the N, P, K, S, Mg, Ca, Fe, Zn and Cl concentrations of the NDNE and the LF that would be added to the system were analysed. Next, the concentrations of LF, NDNE, and RW at the start of the growing season in the simulated system were determined using a non-linear solver technique. This was possible due to the following constrictions:All fractions of LF, NDNE, and RW are greater than zero;The sum of the fractions of LF, NDNE and RW equals 100%;The total N and P contents of the final mixture are below the discharge limits in Flanders (Belgium) [37];The N/P ratio of the medium equals 7.3, as this is the ratio between the N removal and P removal determined in an outdoor duckweed system, with diluted NDNE as the growing medium [19]

The most important restrictions are that the N and P concentrations of the effluent must not exceed the discharge limits in Belgium (resp. 15 and 2 mg/L) [37] and that the N and P removal of the system has to be the same as the N and P addition.

For the preparation of the different growth media, demineralized water (DW) was used instead of rainwater. The results of the non-linear solver technique, and thus the composition of the growing medium at the start of the growing season in the simulated continuous system, are given in Table 1.

Next, the non-linear solver technique was used to calculate the accumulation coefficients of N, P, K, S, Mg, Ca, Fe, Zn and Cl. With these coefficients, it was possible to determine the concentrations of these elements in the growth medium of the system after one growing season, at a variable depth. For these calculations, it was assumed that the overall removal of the different nutrients by the system will be the same as the mean overall removal measured by Devlamynck et al. [19]. The calculated concentrations of the growth media are shown in Table 2.

Duckweed was grown on five different growth media. A reference medium (Table A1), a starting medium and three growing media with the same composition as at the end of the growing season depending on the depth of the tank. These last three growth media were prepared by adding extra nutrients via a specific salt solution in order to obtain the concentrations as shown in Table 2.

All growth media were prepared in volumes of 15 kg. Thus, for the starting medium, 15 g LF was mixed with 235 g NDNE and 14.749 kg DW. For the other growth media, 1 kg of the mass of the DW was replaced by 1 kg of a specific salt solution (Table A4). Afterwards, these growth media were divided into volumes of 3 kg for the tray experiment.

Cl^−^ was added by using NaCl and HCl. HCl was added via titration until a pH equal to that of the starting medium (8–8.2) was obtained. The necessary Cl^−^ concentration was then obtained by adding NaCl.

##### Tray Experiment 2: Different Dilutions/Treatments of NDNE

In the second tray experiment, it was determined to what extent NDNE is a suitable medium for duckweed cultivation. A dose-response curve with nine different growing media was set up (Table A2): (i) a reference medium (EC of 1.5 mS/cm); (ii, iii, and iv) NDNE diluted with demineralized water until an EC equal to 4, 6 and 8 mS/cm; and (v) undiluted NDNE (EC of 9.8 mS/cm).

In reality, the EC can even exceed 9.8 mS/cm. Therefore, these situations were mimicked by (vi and vii) spiking NDNE with a salt solution containing 237.4 mmol/L Na_2_SO_4_, 923.5 mmol/l KCl and 128 mmol/L NaCl until an EC of 11.4 and 11.8 mS/cm; and secondly by (viii and ix) heating NDNE until enough water was evaporated to reach an EC of 11.4 and 11.8 mS/cm.

##### Tray Experiment 3: Salt Tolerance Experiment

Two dose-response tests were conducted on synthetic media. For the first, only NaCl was added to the synthetic medium. In the second, the salt composition of NDNE was mimicked by adding a combination of salts in order to have a Cl/SO_4_ ratio of 4:1 which is similar to the ratio found in the NDNE from previous analyses. 

Therefore NaCl, Na_2_SO_4_ and K_2_SO_4_ were added as Na^+^, K^+^, Cl^−^, SO_4_^2−^ with a molar ratio of 5:1:4:1 (Table 3). The amount of Na^+^ ions were equal for both experiments. 

For both experiments, the pH was corrected by the addition of NaOH. Hence, this caused the addition of an amount of additional Na^+^ in the different treatments; nevertheless, this was considered negligible. Water and biomass samples were taken before inoculation (t0), at the end of week 1 (t1) (pre-cultivation) and at the end of week 2 (t2) (cultivation).

#### 3.1.3. Experimental Duration

The OECD guidelines describe a test period of 1 week as sufficient for toxicity tests [38]. However, phenomena such as luxury consumption have been described before [39]. As a result, a ‘lag phase’ in response curves might occur. Therefore a pre-cultivation period of 1 to 2 weeks was conducted on the same growing media to make sure that duckweed was adapted to the different conditions before the measurements were made. For experiments 2 and 3, the cultivation on the different growth media lasted for 1 week. However, for experiment 1, the cultivation period lasted for 3 days to prevent the depletion of N and P levels in the media.

#### 3.1.4. Analytical Methods

##### Plant FW and DW Determination

Harvested duckweed material was measured both in terms of fresh (FW) and dry weight (DW). First, harvested fresh plant material was rinsed with tap water and drip-dried for 5 min in a fine mesh fishing net. Hereafter, the duckweed mass in the net was dried 2 times with a 5-folded paper towel for approx. 10 min. Afterwards, the duckweed pack was transferred from the net to aluminium cups and weighed on a balance (LA 320P, Sartorius Lab Instruments, Göttingen, Germany). Once the FW was determined and biomass for chlorophyll determination was separated, the samples in the aluminium cups were put in a drying oven at a low temperature (~60 °C) for a minimum of 3 days.

##### Plant Chlorophyll Content

To assess oxidative damage in the salt treatments, chlorophyll content was measured as an indicator via an ethanol extraction, as done by Liu et al. [40]. Therefore, 0.8 g FW was subsequently transferred into 40 mL of a 95% ethanol solution, stored for 5 days in the dark at room temperature, and centrifuged at 2790 rpm for 10 min (Centrifuge 5804 R, Eppendorf, Belgium) where after the supernatant was analysed with a UV-VIS spectrophotometer at 663 and 645 nm (Uvikon XL, Biotek Instruments, Santa Clara, CA, USA) to obtain chlorophyll contents. The concentrations of chlorophyll were calculated according to Huang et al. [41]:(1)$$C_{a}\;\left[\frac{mg}{L}\right]=12.72A_{663}+2.69A_{645}$$
(2)$$C_{b}\;\left[\frac{mg}{L}\right]=22.90A_{645}-4.68A_{663}$$
(3)$$C_{chl}\;\left[\frac{mg}{L}\right]=C_{a}+C_{b}$$
where *C_a_*, *C_b_* and *C_chl_* represent the content of chlorophyll a, chlorophyll b, and total chlorophyll, respectively; *A*_663_ and *A*_645_ are the absorbances at 663 and 645 nm, respectively.

##### Compositional Analysis of Duckweed

The total N content (T-N) of the duckweed was determined before and after cultivation. A CN analyser (Primacs SNC-100, Skalar, Breda, The Netherlands) was used to determine the total nitrogen content in the duckweed. With the T-N content, the N removal and uptake of the plant were calculated.

For plant Ca, Mg, Na, K, P, S, Al, Cu, Fe, Mn and Zn content before and after cultivation, dried plant material, to which 65% HNO_3_ was added, was first digested in a microwave (Milstone Ultrawave, SRC technology). Next, samples were accordingly diluted prior to elemental determination with inductively coupled plasma-optical emission spectrometry (ICP-OES) (Vista-MPX, Varian Inc., Palo Alto, CA, USA).

##### Analysis of NDNE, LF and Growing Media

For the determination of the T-N content of the undiluted NDNE and LF, the same CN analyser was used. However, as it was expected that the liquid samples contained nitrate and ammonia, 4 times the sample weight of sucrose and 1–2 drops of 20% o-Phosphoric acid solution were added to the crucibles before T-N analysis to maximize the N yield from NO_3_^−^ and NH_4_^+^ in the sample.

For the determination of Ca, Mg, Na, K, P, S, Cu, Fe, Mn and Zn concentrations in the waste streams and the more diluted growth media before and after cultivation, the samples were first digested on a hot plate with a solution of H_2_O_2_ and HNO_3_ (65%) in a 1:2 ratio for 30 min. After this, the sample was checked for its transparency. If the sample was still not transparent, then H_2_O_2_ and HNO_3_ were again added at the same ratio used to continue the digestion until a transparent sample was obtained. Then the sample was filtered through a Whatman, grade 5 filter paper and diluted with milli-Q. Next, samples were accordingly diluted prior to elemental determination with ICP-OES.

The determination of Cl^−^, NO_3_^−^, PO_4_^3−^, SO_4_^2−^ in the water samples was done using ion chromatography (761 Compact Ion Chromatograph, Methrom, Herisau, Switzerland), preceded by 0.45 µm syringe filtration and dilution.

Finally, pH and electric conductivity (EC) was measured with a pH-meter (ProfiLine pH 3110, WTW, Weilheim, Germany) and a conductivity tester, respectively (ProfiLine Cond 3110, WTW, Weilheim, Germany).

### 3.2. In Situ Monitoring of NDNE

At the pig farm of IVACO, Eernegem, Belgium, pig manure is treated by a combination of centrifugation and subsequent biological treatment (Trevi, Ghent, Belgium) of the liquid fraction (LF). The loading rate is 20 m³/day and the total retention time is 36 days. LF has a higher content of N and a higher N/P ratio than the solid fraction. NDNE is the result of the biological treatment process and was monitored from 26 November 2019, until 28 June 2020. For this, a solar power-driven 3798-S digital inductive electrical conductivity sensor (Hach, Belgium) was installed in the aeration tank in the last step of the process, as shown in Figure 1.

In order to assess the influence of weather conditions on conductivity, the monitored EC values were compared with climatological data obtained from the Royal Meteorological Institute. The climatological data also ran from 26 November 2019, to 28 June 2020. It was calculated how much water entered the manure treatment through precipitation. For this, the retention time of the system (36 days) and that of the different tanks individually were taken into account. For example, when one drop of rainwater enters the first step/tank of the treatment process, it will stay in the process until it leaves the last tank in the treatment (at day 36). Taking this into account, the cumulative precipitation was determined.

The biological manure treatment system of IVACO is divided into 4 tanks, 2 anaerobic and aerobic. For each tank, the cumulative precipitation was calculated as follows:(4)$$CP_{k}=\sum_{i=0}^{n}P_{i}*\;\frac{t_{n}-i}{t_{n}},$$
with $CP_{k}$ as the cumulative precipitation for tank *k*, $P_{i}$ the precipitation on day i and with n the retention time (in days) of the specific tank. Next, the total cumulative precipitation for the whole system was calculated as follows:(5)$$CP_{tot}=\sum_{k=1}^{4}CP_{k}*\;\frac{A_{k}}{A_{tot}},$$
with $A_{k}$ as the surface area of tank k of the system and $A_{tot}$ the total surface area of the system. With this total cumulative precipitation, the corrected EC was calculated as follows:(6)$$EC_{cor}=\frac{EC_{measured}}{1-\left(\frac{\frac{CP_{tot}}{1000}*A_{tot}}{V_{tot}}\right)\;},$$
with $A_{k}$ and $V_{tot}$ the total surface area and volume of the manure treatment system.

### 3.3. Calculations and Statistics

The relative growth rate was calculated based on the dry weight as follows [42]:(7)$$RGR\;\left[g/g/d\right]=\frac{\ln\left(DW_{e}\right)-\ln\left(DW_{i}\right)}{t_{2-}t_{1}},$$
with *DW_e_* and *DW_i_* representing respectively the dry weight of duckweed after (*t*_2_) and before (*t*_1_) cultivation. To make it possible to compare data from different experiments, the relative growth rate inhibition was also calculated:(8)$$RGRinhibition_{i}\;\left[\%\right]=1-\frac{RGR_{i}}{\mathrm{mean}\left(RGR_{ref}\right)},$$
with *RGR_ref_* representing the RGR of the reference medium of the experiment. For experiment 1, where a continuous system was simulated, the start medium was chosen as the reference RGR. For experiments 2 and 3, the synthetic medium without the addition of extra salts was chosen as the reference RGR. 

The same was done for the total chlorophyll concentration, the relative chlorophyll concentration was calculated as follows:(9)$$Chlorophyll\;inhibition_{i}\;\left[\%\right]=1-\frac{C_{Chl\;i}}{\mathrm{mean}\left(C_{Chl\;ref}\right)}$$
with *C_Chl ref_*, representing the total chlorophyll concentration of the reference medium of the experiment. The same references were chosen as for the calculation of the RGR inhibition.

Nutrient uptake/recovery rates were calculated considering the dry weight gain as well as the change in nutrient content in the biomass as follows:(10)$$Nutrient\;uptake\;\left[g/m^{2}/d\right]=\frac{C_{i}*DW_{i}-C_{0}*DW_{0}}{area*\;t_{2}-t_{1}}$$
with *c*_0_ and *c_i_* representing respectively the content of the specific nutrient in the plant before and after cultivation. For these concentrations, a correction was always made for the amount of water that evaporated during the cultivation step.

Nutrient removal by the system was calculated considering the nutrient concentrations of the growing media before and after cultivation as follows:(11)$$Nutrient\;removal\;\left[g/m^{2}/d\right]=\frac{c_{0}-c_{i}}{area*\left(t_{2}-t_{1}\right)\;},$$
with *c*_0_ and *c_i_* representing respectively the content of the specific nutrient in the plant before and after cultivation. These concentrations were also corrected for the amount of water that evaporated during the cultivation step.

Microsoft Excel and R Statistical Software (v3.6.1, R Core Team 2019, Vienna, Austria) were used for statistical data processing and visual display. To show significant differences between the treatments in the experiments, parametric analyses such as one-way ANOVA, two-way ANOVA and Tukey’s HSD tests were performed. These tests were only used if all conditions were met. The condition of whether the residuals are normally distributed or not was checked graphically by constructing qq-plots and numerically via the Shapiro-Wilk test. Homoscedasticity of the variations was checked graphically using box plots and numerically using the Modified Levene’s-Test. If these conditions for these parametric analyses were not met, significant differences were demonstrated using the Kurskal-Wallis test followed by a post hoc Dunn’s-Test. A significance value of 5% was used for all analyses, and in all cases the sample size (n) was equal to four.

## 4. Conclusions

This study showed that a continuous duckweed cultivation system can be used for the treatment of NDNE and the simultaneous provision of protein-rich biomass on a large scale. It was shown, however, that it is important to take into account that salts accumulate in long operating systems when their concentration added via the waste stream is higher than the concentration that can be removed by the system. This study concluded that, when duckweed is grown in a system with a continuous supply of NDNE and LF and a continuous discharge that meets the legal standards in Flanders (Belgium) (15 mg N/L and 2 mg P/L), a buffering capacity of at least 1 m depth is needed. This would ensure that the relative growth would not be inhibited at the end of the growing season compared to the start of the cultivation. However, in that case, this stand-alone treatment system would consume around 5 to 10 times more water than other crops such as maize. On the other hand, it was observed that duckweed grew better on NDNE (till 75% NDNE, or EC of 8 mS/cm) than on the reference N-medium, indicating that a lower dilution rate might be used if the duckweed system would be connected to a wetland for reaching the dischargeable legal limits. This study showed that the ratio of anions can partly explain this higher salt tolerance when grown on NDNE, but other mechanics remain uncovered. It is suggested that organic substances might have an effect on reducing salt stress. This observation increases the potential of using pig manure waste streams for duckweed cultivation. However, it was demonstrated by in-situ monitoring of the NDNE of a manure treatment plant that the composition of NDNE is not constant over time. Therefore, the optimal dilution found in the tray test is only temporarily valid. Further research on the operation of such a system, with the accumulation of elements taken into account, was shown to be relevant for the future application of this technology.

## Figures and Tables

**Figure 1 plants-11-03189-f001:**
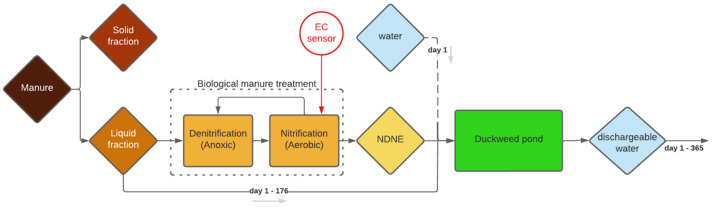
Schematic representation of the manure treatment process with centrifugal separation as the first step, biological treatment as the second step (with nitrification-denitrification = NDN) and a duckweed pond as the third and final treatment step. The duckweed pond in this scheme is not yet applied in practice, the other treatment steps are presented in the same order as they occur at IVACO, a pig manure treatment plant in Flanders (Belgium), Eernegem.

**Figure 2 plants-11-03189-f002:**
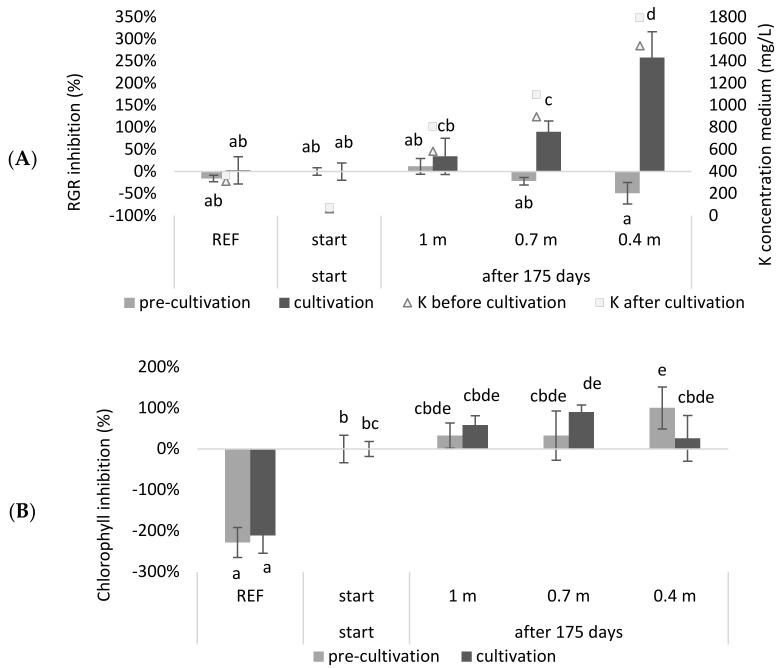
The relative growth rate inhibition (**A**), and the chlorophyll inhibition (**B**) of duckweed cultivated on a simulation of the concentration of wastewater of a continuous system, before and after 175 growing days, with different buffering capacities. Significant differences (n = 4 and *p* < 0.05) per graph are indicated by different letters. Error bars indicate standard deviations.

**Figure 3 plants-11-03189-f003:**
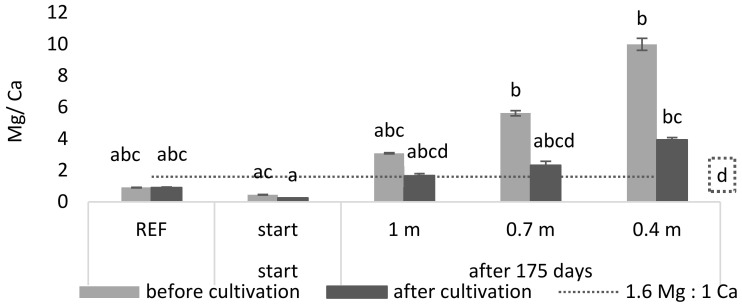
The Mg/Ca ratio in the medium of the simulated continuous system, before and after 175 growing days, with different buffering capacities. The dashed line indicates the minimum permitted Mg concentration in relation to the Ca concentration in the medium. Significant differences (n = 4 and *p* < 0.05) per graph are indicated by different letters. The letter in the dashed square belongs to the dashed line. Error bars indicate standard deviations.

**Figure 4 plants-11-03189-f004:**
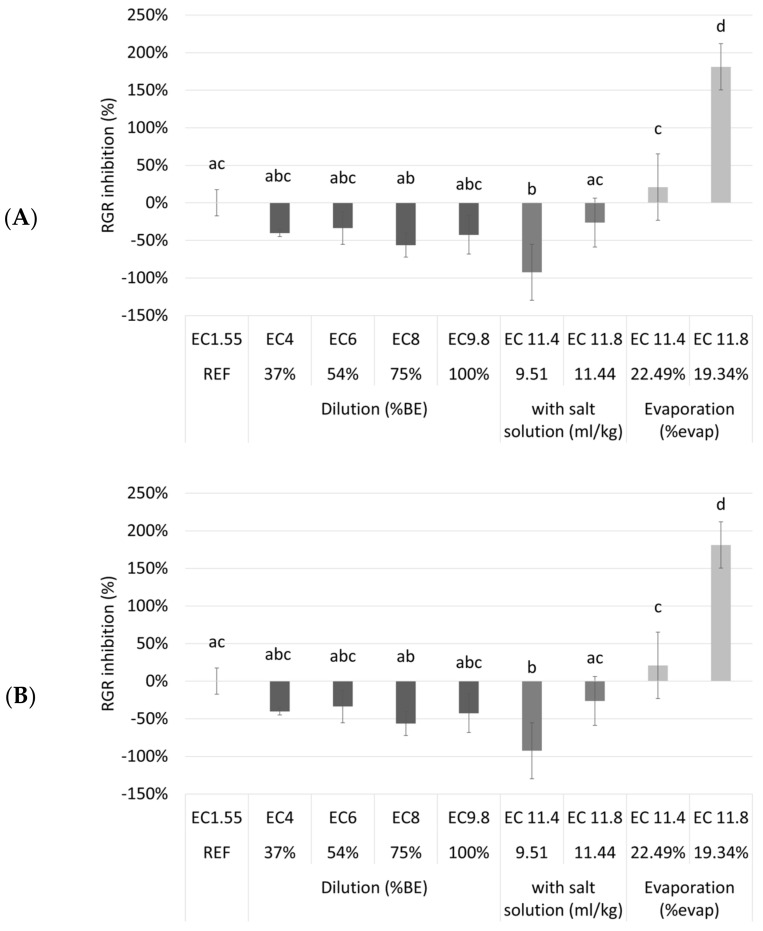
The relative growth rate inhibition (**A**) and the chlorophyll inhibition (**B**) of duckweed cultivated in different dilutions of NDNE, and NDNE to which a salt solution was added and in NDNE ’concentrated’ after evaporation. On the x-axis, the EC values and respectively the percentage of dilution, the concentration of salt solution or the percentage of evaporation are given. Significant differences (n = 4 and *p* < 0.05) per graph are indicated by different letters. Error bars indicate standard deviations.

**Figure 5 plants-11-03189-f005:**
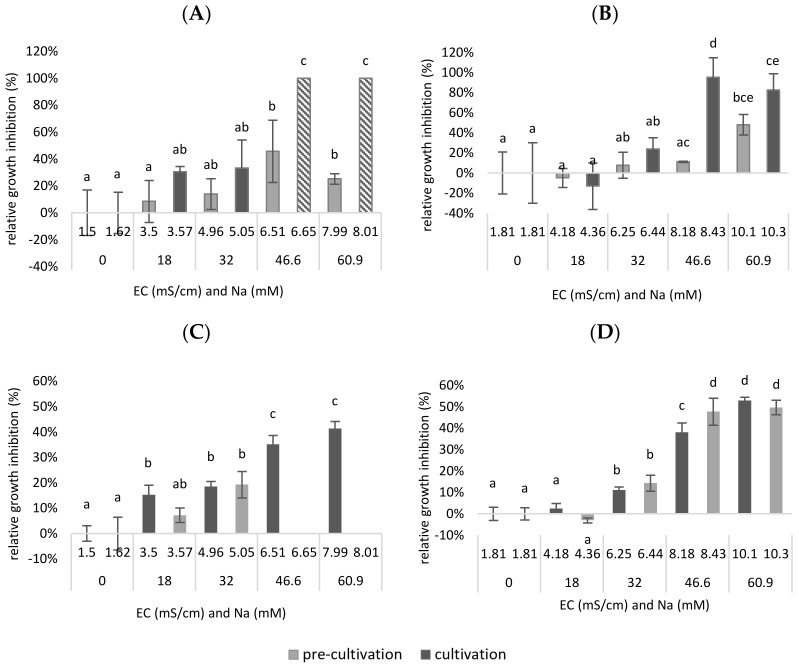
The relative growth rate inhibition of duckweed cultivated on a synthetic medium after NaCl addition (**A**) or NaCl, Na_2_SO_4_ and K_2_SO_4_ addition (**B**); the chlorophyll inhibition of duckweed cultivated on a synthetic medium after NaCl addition (**C**); or NaCl, Na_2_SO_4_ and K_2_SO_4_ addition (**D**). On the x-axis both the EC and the concentration of sodium in the medium is shown. A distinction is made between pre-cultivation and cultivation. For the inhibitions of the first experiment (**A**,**C**) there was no data obtained for the duckweed during cultivation grown on a medium with 46.6 and 60.9 mM Na due to die-off; therefore, in the RGRinh graph, a default value of 100% inhibition was taken (= no growth). Significant differences (n = 4 and *p* < 0.05) per graph are indicated by different letters. Error bars indicate standard deviations.

**Figure 6 plants-11-03189-f006:**
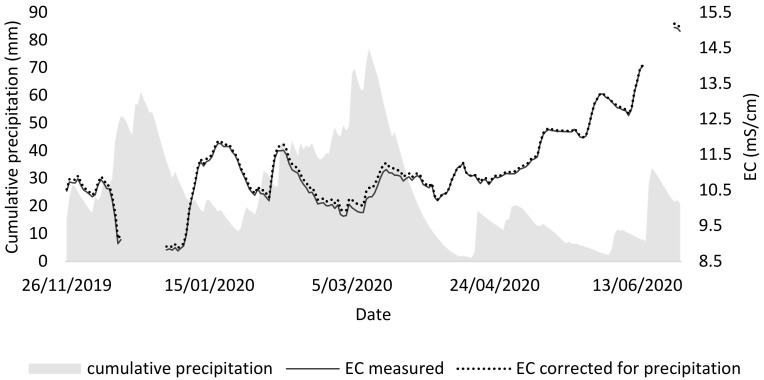
Electric conductivity (EC) measured in the aerobic tank (last tank) of a biological pig manure treatment system, together with the cumulative precipitation that could enter the system over a retention time of 36 days. Additionally, a corrected EC was calculated for the dilution from precipitation.

**Figure 7 plants-11-03189-f007:**
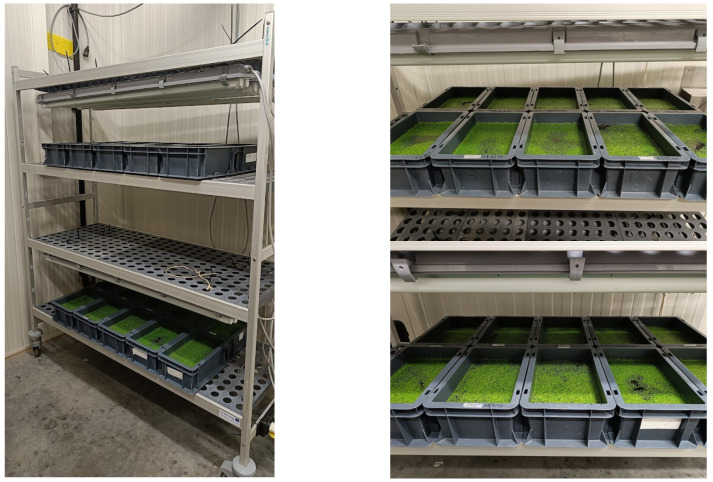
Picture of the growth rack used for all tray experiments, with 10 PET containers and 4 parallel TL-lights per level.

**Table 1 plants-11-03189-t001:** The total N, total P and the N to P ratio of the LF, NDNE and DW, together with the calculated composition of the mixture after a non-linear solver technique maximising the NDNE composition within presented restrictions.

	Total N (mg/kg)	Total P (mg/kg)	N:P Ratio	Mass Fraction (%)
LF	3440	191		0.1
NDNE	725	115		1.6
DW	0	0		98.3
Mixture	14.9	2	7.4	100
Restrictions	15	2	7.4	100

**Table 2 plants-11-03189-t002:** The simulated nutrient concentrations of the different growth media were calculated using a non-linear solver technique. ‘Start’ = the composition of the growing medium of a continuous system at the start of the growing season; ‘1 m’/‘0.7 m’/‘0.4 m’ = the composition of the growing medium of a continuous system after 175 days of cultivation in a 1 m/0.7 m/0.4 m deep tank.

	Start	After One Growing Season		
Nutrient (mg/L)	(All Depths)	1 m	0.7 m	0.4 m
N	14.86	16.82	17.65	19.75
P	2.03	2.75	3.06	3.83
K	60.15	839.63	1173.70	2008.86
T-S	5.14	64.65	90.15	153.90
Mg	0.76	5.19	7.09	11.84
Ca	3.16	2.49	2.20	1.47
Fe	0.78	3.29	4.37	7.06
Zn	0.17	2.04	2.84	4.85
Cl	39.11	554.12	774.85	1326.65

**Table 3 plants-11-03189-t003:** Salt concentrations added to the synthetic N medium during the first two salt experiments. The EC values were temperature corrected, and the interval gives the maximal range of measured values (at t0 and t1). Treatment n°1 of both experiments served as a reference.

	N°	NaCl (mM)	Na_2_SO_4_ (mM)	K_2_SO_4_ (mM)	EC (mS/cm)
Treatment NaCl	1	0.0	-	-	1.6–1.6
	2	18	-	-	3.5–3.6
	3	32	-	-	5.0–5.1
	4	46.6	-	-	6.5–6.7
	5	60.9	-	-	8.0–8.0
Treatment Cl/SO_4_ (Ratio 4:1)	1	0.0	-	-	1.8–1.8
	2	14.4	1.8	1.8	4.2–4.4
	3	25.6	3.2	3.2	6.3–6.4
	4	37.3	4.7	4.7	8.2–8.4
	5	48.7	6.1	6.1	10.1–10.3

## Data Availability

The data presented in this study are available on request from the corresponding author.

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
