# Peer review of "The Impact of Salt Accumulation on the Growth of Duckweed in a Continuous System for Pig Manure Treatment"

_plants, 2022, doi:10.3390/plants11233189_

Round 1
Reviewer 1 Report
A serious environmental problem, it is good that the Authors have taken up this research topic.
The paper requires minor changes. Here are my comments:
1. You use in Your article abbrevations "NDNE" and "NDN" with a word "effluent" which is a little bit confusing. I suggest use only "NDNE"
2. It is enough to explain one time "NDNE" abbrevation - you do not need to do this again (Table 1)
3. Line 139 - Fig 1 and line 408 Fig 1? You need to renumber of figures
4. The text from lines 160-171 does not fit Materials/methods chapter
5. Try to replace this expression "..it would be intersting..." line 395, 643 by more scientic expalantion about the reasons for starting the research
6. Lines 384-394 - this paragraph should be rather in Introduction chapter
7. Lines 440-444 - I do not see why you put this text here
8. Line 543 "In NDNE there is more nitrogen present in the form of ammonium, which is the 543 preferred nitrogen source of duckweed" - I could find this data
9. Chapter 3.2.3 Variations in NDNE and lines 662-663: "However, it was demonstrated by in-situ monitoring of the NDNE of a manure treatment plant that the composition of NDNE is not constant over time" . It is obvious that the NDNE concentration is changing, I do not know if this statement should be included in the conclusions and that a whole chapter must be devoted to it.
10. The paper is too long, please try to shorten it.
Reviewer 2 Report
The authors tested the ability of duckweed to treat nitrification-denitrification effluents from pig manure. They run three different experiments in order to design a stand-alone system and to investigate the salts concentration at which the growth rate of the plants is inhibited. The authors identified a depth and a volume of water that should be added to the system in order to prevent growth inhibition and to discharge effluents compliant to the Belgian legislation. The results of this study suggest that a stand-alone system requires the addition of a significant amount of water. The authors concluded that the pig manure should be treated by a system that couples duckweed and constructed wetland.
The study is interesting, novel, very useful and well designed, however the quality of the manuscript should be improved. I personally struggled a bit to go through the text and I believe it is not very reading-friendly, especially for readers not familiar with this type of effluents. The structure of some sections should be reorganized and the authors should make sure to explain evry term and concept they mention. Please see below some specific comments.
Abstract
line 24 Flanders. Where is Flanders? I now know it's in Belgium, but this is what I thought when I first read the abstract
Lines 25-27 the study is not described properly. I could not understand what your study was about from your abstract.
Introduction
Line 40. Flanders, again where is it?
lines 41-42. Why nutrients become unaviable for crop production? It's unclear why you are saying this
Line 45. Duckweed can be seen as a floating wetland? I would change with: duckweed form a floating mat
Line 50. I think the international duckweed community agrees that duckweed belong to the family of Lemnaceae. See: Tippery, N. P., Les, D. H., Appenroth, K. J., Sree, K. S., Crawford, D. J., & Bog, M. (2021). Lemnaceae and Orontiaceae are phylogenetically and morphologically distinct from Araceae. Plants, 10(12), 2639.
Line 53. Threre are more recent studies that assess duckweed productivity in northern Europe. e.g. Paolacci, S., Stejskal, V., Toner, D., & Jansen, M. A. (2022). Wastewater valorisation in an integrated multitrophic aquaculture system; assessing nutrient removal and biomass production by duckweed species. Environmental Pollution, 302, 119059.
Line 61. Also very old citation. Please site: Stejskal, V., Paolacci, S., Toner, D., & Jansen, M. A. (2022). A novel multitrophic concept for the cultivation of fish and duckweed: A technical note. Journal of Cleaner Production, 366, 132881.
Material and methods
line 86 I believe it's m not m3
Lines 89-91 The design is uclear, maybe add a pic or a schematic representation?
Line 98. What's a reference tray? explain it now
Lines 137-148. This is a very nice figure that explains very well the manure treatment process, but I would move it to the introduction section. This should be one of the first things you explain.
Lines 152 - 158 this also shoudn't be in M&M section
Tray experiments 2 and 3 should be explained better
Please provide a reference for the formulas you use: (6), (7), (8), (9), (10) and (11)
Results and and discussion
The results are well discussed, but, again, the structure of this section is a bit chaotic. Start each subsection with the actual results and only after displaying the results, discuss them.
As last consideration, this study didn't test the quality of the biomass of duckweed produced. Osmotic stress can cause a reduction in protein concentration. You should at least mention in the discussion that the next step to this study should be biomass quality assessment.
Round 2
Reviewer 2 Report
Dear authors,
the quality of the manuscript is highly improved and my opinion is that the paper can now be published. I only have a small content: line 63 states that reference n.8 refers to the south of Ireland. Please remove 'south of', the paper describes a system located in the middle of Ireland.
Thank you